# A Novel Method for Extrinsic Calibration of Multiple RGB-D Cameras Using Descriptor-Based Patterns

**DOI:** 10.3390/s19020349

**Published:** 2019-01-16

**Authors:** Hang Liu, Hengyu Li, Xiahua Liu, Jun Luo, Shaorong Xie, Yu Sun

**Affiliations:** 1School of Mechatronic Engineering and Automation, Shanghai University, No. 99 Shangda Road BaoShan District, Shanghai 200444, China; scholar.hang@gmail.com (H.L.); shuluojun@126.com (J.L.); shusrxie@126.com (S.X.); sun@mie.utoronto.ca (Y.S.); 2School of Mechanical Engineering, Beijing Institute of Technology, Beijing 100081, China; liuxiahua@bit.edu.cn; 3Department of Mechanical and Industrial Engineering, University of Toronto, Toronto, ON M5S 3G8, Canada

**Keywords:** camera calibration, depth sensors, 3D vision, omnidirectional vision

## Abstract

This paper presents a novel method to estimate the relative poses between RGB-D cameras with minimal overlapping fields of view. This calibration problem is relevant to applications such as indoor 3D mapping and robot navigation that can benefit from a wider field of view using multiple RGB-D cameras. The proposed approach relies on descriptor-based patterns to provide well-matched 2D keypoints in the case of a minimal overlapping field of view between cameras. Integrating the matched 2D keypoints with corresponding depth values, a set of 3D matched keypoints are constructed to calibrate multiple RGB-D cameras. Experiments validated the accuracy and efficiency of the proposed calibration approach.

## 1. Introduction

In recent years, low-cost and efficient depth and color (RGB-D) devices such as the Microsoft Kinect, Intel RealSense, and Structure Sensor have attracted much attention because of their applications in indoor scene reconstruction and robot navigation. The depth cameras of these devices can provide a depth map with a VGA resolution (640 × 480) at video rate (e.g., 30 Hz) using efficient light-coding technologies that avoid the challenging task of dense 3D reconstruction from color images. The 3D models reconstructed using these depth cameras have been used to generate more realistic 3D content for virtual reality (VR) [1] and help align the rendered virtual objects with real scenes for augmented reality (AR) [2]. Furthermore, the direct depth sensing capability of these depth cameras is particularly suitable for robots to navigate in an unknown environment.

With an RGB-D camera, the simultaneous localization and mapping (SLAM)-based approach is mainly used for fusing point cloud frames to reconstruct indoor scenes [3,4,5,6]. However, hundreds or even thousands of frames must be captured in state-of-the-art SLAM systems to reconstruct a common indoor environment, such as a room or an office [7] because of two problems. (1) The field of view (FoV) of depth cameras is limited; thus, only a small part of the scene is represented in a single frame. The Kinect, for example, has a horizontal FoV of 57${}^{\circ }$, which is much smaller than the horizontal 240${}^{\circ }$ FoV of the Hokuyo URG-04LX-UG01, a laser scanner with a similar range and measurement accuracy to the Kinect [8]. (2) To track the poses of depth cameras to effectively fuse multiple point cloud frames, consecutive frames must be captured to have sufficient scene overlap. Typically, more than ninety percent of overlap is required, which further increases the number of frames for reconstruction.

One solution to these problems is to use a multi-camera setup in which RGB-D cameras face different directions to sample different sections of the environment [9,10,11,12]. However, complications can occur with the use of multiple cameras, such as more difficult calibration caused by the minimal overlapping FoV between cameras.

Classical extrinsic calibration strategies such as the chessboard-based method and the keypoints-based method cannot be applied to calibrate RGB-D cameras in a multi-camera setup, because their overlap requirement constitutes a very strong constraint. Martyna Poreba et al. proposed a linear feature-based method to perform accurate registration of point clouds [13]. Fernandez-Moral et al. proposed to calibrate multiple RGB-D cameras through planes or lines [9,14]. Planes and lines have large spatial spans; thus, they can be observed by cameras with little or no overlapping FoV.

Another kind of approaches that is proposed to calibrate multiple RGB-D cameras is based on per-camera odometry [15,16]. Cameras are calibrated by finding all camera odometry transforms-based on matched features from frames that are captured in the motion paths of all cameras. In these methods, SLAM or visual odometry techniques are applied to estimate camera trajectories. However, the robustness of SLAM and visual odometry techniques highly depends on the environment.

In this work, a new extrinsic calibration method that relies on descriptor-based patterns is proposed to estimate the relative poses between the RGB-D cameras with minimal overlapping FoV. The descriptor-based pattern is based on the reverse engineering of SIFT/SURF, can provide many more well-matched keypoints than nature scenes [17]. In our method, a set of matched 3D points are constructed based on extracted 2D keypoints and corresponding depth values from depth maps. The relative poses between RGB-D cameras are estimated efficiently by minimizing the alignment error of matched 3D points. If there are loop closures in a multi-RGB-D camera system, the estimated poses can be further optimized globally to decrease incremental estimation error through pose graph optimization. An efficient distributed capturing system is also proposed to provide multiple RGB-D data frames in real time to facilitate the usage of multi-RGB-D cameras system in practical applications such as 3D reconstruction and robot navigation.

## 2. Method Overview

Figure 1 shows the flow chart of our calibration method. The input to the method is color images and depth maps (RGB-D frames) captured in a scene with descriptor-based patterns. The descriptor-based patterns [17] are random textures generated using the reverse engineering of SIFT/SURF, can provide many more well-matched keypoints than nature scenes even in the case of minimal overlapping FoV between cameras. The SIFT keypoint detector is used in this work, as it is more stable to rotation and illumination changes than SURF [18].

Next the well-matched keypoints are extracted from color images of descriptor-based patterns using the RANSAC (random sample consensus)-based method. Then 3D point sets are constructed based on the extracted 2D keypoints and their depth values in the depth maps.

Afterwards the relative poses are estimated by directly minimizing the alignment error of 3D point sets. The last step is to optimize the estimated poses with possible loop closure, for example the first and the last RGB-D camera in a panoramic setup (see Figure 2a) have an overlapping FoV that can be seen as a loop closure. Since this is a conditional step, the last step is presented within a dashed box.

## 3. Detailed Methods

### 3.1. Keypoint Extraction

The descriptor-based calibration pattern is composed of several random noise images at different scales in accordance with the mechanism of SIFT/SURF. Compared with natural scenes, this pattern contains a high number of detectable features on multiple scales that can be easily detected by a camera at varying distances. Thus, the descriptor-based pattern can provide many and more accurately matched keypoints between two cameras in the case of minimal overlapping FoV.

In Figure 3, the detected keypoints are represented by green dots. Figure 3a,b shows the detected keypoints of the image captured in a natural scene, and the image captured in a scene with descriptor-based patterns, respectively. There are only 406 keypoints in Figure 3a, but 1231 keypoints in Figure 3b. Figure 4a,b shows the matched keypoints of a pair of images captured by two cameras with an approximately thirty percent overlapping FoV. Well-matched keypoints are connected by green lines; poorly matched keypoints are connected by red lines. The matching results are generated by the Flann-based descriptor matcher implemented in OpenCV, where matched keypoints are accepted only if its descriptor distance is less than three times the minimum descriptor distances. There are 56 well-matched keypoints and 32 poorly matched keypoints in Figure 4a, 116 well-matched keypoints and 30 poorly matched keypoints in Figure 4b. Thus the descriptor-based patterns provide more robust matched keypoints.

To further increase the efficiency and accuracy of extracting the matched keypoints, the keypoint detector can be only applied to the overlapping regions of neighboring images, and a RANSAC-based filtering method is applied to refine matches. The overlapping regions can be roughly estimated based on the FoV of cameras and the angle between two neighboring cameras. In the RANSAC-based filtering method, RANSAC is used to select the best subset of corresponding keypoint pairs through estimating the homography matrix. A re-projection error threshold is predetermined to help the RANSAC-based filtering method to distinguish inliers from outliers.

Figure 4c shows the matched keypoints when the keypoints are only detected in the regions within the blue rectangles (overlapping regions). There are 362 well-matched keypoints in Figure 4c, no obviously poorly matched keypoints. Figure 4d shows the selected 25 keypoint matches after applying the RANSAC-based method, the re-projection error threshold is set to 0.3 pixel.

### 3.2. Estimate the Pose from 3D Point Sets

Based on these matched keypoints, we find their corresponding depth values from the depth maps generated by the depth camera in the RGB-D camera to construct 3D point sets to estimate poses. This is a class of ICP (iterative closest point) problem that minimizes the difference between two point clouds. The critical difference is that correspondences between two point clouds are already known in our case, thus the time-consuming finding of matched 3D poits in typical ICP-based method can be avoided.

We construct a non-linear least squares formulation based on the alignment error of matched 3D points that can be solved by the Levenberg–Marquardt algorithm (LM) considering the error of selected matched keypoints and depth measurement. When the correspondences between two point clouds are known, there exists an analytic solution for this formulation [19] which guarantees the fast convergence of LM without the good initial guess. Implementation details are as follows.

#### 3.2.1. Align the Depth Map with the Color Image

Due to the different spatial positions and intrinsic parameters of the depth camera and of the color camera in a RGB-D camera, the depth map is not aligned with the color image. The extrinsic parameters between the depth and color camera, intrinsic parameters of depth and color camera are used to align the depth map with the color image. SDKs (software development kits) of RGB-D cameras provide interfaces to obtain the intrinsic parameters and to align the depth map with color image. Since the users may want to obtain a more precise alignment result through calibrating the intrinsic and extrinsic parameters by themselves, the detailed methods are provided.

Let $\left(u_{0},v_{0}\right)$ denote the coordinates of the principal point of the depth camera, $f_{x}$ and $f_{y}$ denote the scale factors in image *u* and *v* axes of the depth camera, and $u_{0}$, $v_{0}$, $f_{x}$ and $f_{y}$ be the intrinsic parameters of the depth camera. Let [u,v,Z] represent a pixel in the depth map, *Z* represent the depth value in [u,v], and ${\left[X,Y,Z\right]}^{T}$ represent the mapped 3D point in [u,v] in the depth camera coordinate system. According to the pinhole camera model, the values of *X* and *Y* can be calculated according to

(1)$$\begin{array}{c} X=\left(u-u_{0}\right)Z/f_{x},\\ Y=\left(v-v_{0}\right)Z/f_{y}. \end{array}$$

Let *T* represent the extrinsic parameter (i.e., transformation matrix) between the depth and color camera, $\left(u_{0}^{\prime },v_{0}^{\prime }\right)$ denote the coordinates of the principal point of the color camera, ${f_{x}}^{\prime }$ and ${f_{y}}^{\prime }$ denote the scale factors in image $u^{\prime }$ and $v^{\prime }$ axes of the color camera. The aligned depth pixel $\left[u^{\prime },v^{\prime },Z^{\prime }\right]$ is obtained by transforming ${\left[X,Y,Z\right]}^{T}$ based on *T* and get ${\left[X^{\prime },Y^{\prime },Z^{\prime }\right]}^{T}$, then mapping ${\left[X^{\prime },Y^{\prime },Z^{\prime }\right]}^{T}$ to the color image coordinate system as follows (2)$${\left[X^{\prime },Y^{\prime },Z^{\prime }\right]}^{T}=T{\left[X,Y,Z\right]}^{T},\;u^{\prime }=\frac{X^{\prime }}{Z^{\prime }}{f_{x}}^{\prime }+{u_{0}}^{\prime },\;v^{\prime }=\frac{Y^{\prime }}{Z^{\prime }}{f_{y}}^{\prime }+{v_{0}}^{\prime }.$$

#### 3.2.2. Estimate the Pose

Let $\left[u_{j},v_{j}\right]$ and $\left[u_{j}^{\prime },v_{j}^{\prime }\right]$ denote the coordinates of selected matched keypoints, $Z_{j}$ and $Z_{j}^{\prime }$ denote the corresponding aligned depth value in $\left[u_{j},v_{j}\right]$ and $\left[u_{j}^{\prime },v_{j}^{\prime }\right]$ obtained from the aligned depth map respectively, j=1,2,…,N, we construct two 3D point sets $\left\{p_{j}\right\}$, $\left\{p_{j}^{\prime }\right\}$ based on Equation (Equation 1). The relative pose between these two 3D point sets can be found by minimizing
(3)minξ=12∑i=1npj−exp(ξ^)p′j22, where ξ∈se(3) is a vector with six dimensions that represents the camera pose, ξ is mapped to a matrix with four dimensions $R^{4\times 4}$ by ^. $\xi^{\hat{\;}}$ is mapped to T∈SE(3) by the exponential map exp(). We use the standard Levenberg–Marquardt algorithm to solve Equation (Equation 3). The derivative with respect to the camera pose of an error element is given by
(4)∂(exp(ξ^)p)∂ξ=limδξ→0exp(δξ^)exp(ξ^)p−exp(ξ^)pδξ≈limδξ→0(I+δξ^)exp(ξ^)p−exp(ξ^)pδξ=limδξ→0δξ^exp(ξ^)pδξ=limδξ→0−(exp(ξ^)p)ξ^δξδξ=−(exp(ξ^)p)^, where ${}^{\hat{\;}}$ denotes the mapping from $\exp(\xi^{\hat{\;}})p$ represented by ${\left[x,y,z\right]}^{T}$ to the corresponding anti-symmetric matrix $\left[\begin{array}{ccc} 0 & -z & y\\ z & 0 & -x\\ -y & x & 0 \end{array}\right]$.

### 3.3. Pose Graph Optimization

Assume that there are *M* RGB-D cameras in a panoramic 3D vision system. There is an overlapping FoV between each pair of adjacent RGB-D camera. Let $T_{i,j}$ denote the estimated relative poses between two adjacent RGB-D cameras, i=1,2,…,M; j=i+1, when *i* equals *M*, *j* is 1.

When the position of the first RGB-D camera $x_{1}$ is set to ${\left[0,0,0,1\right]}^{T}$, we can calculate the position $x_{i}$ for each RGB-D camera based on $T_{i,j}$. Due to the pose estimation error, the estimated position of the first RGB-D camera that is calculated by $T_{N,1}x_{N}$ is not equal to ${\left[0,0,0,1\right]}^{T}$. This accumulated error can be mitigated by the loop-closure constraint provides by the first and last RGB-D camera. In this case, the pose optimization problem can be seen as a small scale simultaneous localization and mapping (SLAM) problem. Estimated poses can be optimized using the pose graph optimization method in SLAM [20,21]. According to the pose graph optimization theory, the problem can be solved by finding the minimum of a function of this form:(5)$$x^{*}=\arg\;\;\min_{x}\sum_{(i,j)\in S}r_{i,j}^{T}\left(x\right)\Lambda_{i,j}r{}_{i,j}\left(x\right),$$ where $x={\left(x_{1}^{T},\;\ldots ,x_{n}^{T}\right)}^{T}$ is a vector of poses, $r{}_{i,j}$ is the residual of the predicted and observed relative poses between the *i*-th and *j*-th node, $\Lambda_{i,j}$ denotes the measurement information matrix, and *S* represents the set of edges that connect the nodes.

## 4. Experimental Results

### 4.1. System Setup

Figure 2a shows the panoramic RGB-D camera setup that we built to evaluate the proposed calibration method. It consists of 12 Kinect v1 cameras, all Kinects are vertically positioned for a more compact design. The Kinect v1 camera has an angular field of view (FoV) of 43${}^{\circ }$ from the vertical. The overlap FoV of two neighboring cameras is 13${}^{\circ }$$\left(\left(43^{\circ }\times 12-360^{\circ }\right)/12\right)$ which is only approximately 30 percent of the vertical FoV of each camera.

We also designed a distributed system built on top of a local area network (LAN) to capture the RGB-D frames from all twelve cameras in real time. The distributed capturing system (see Figure 2c) consists of twelve Raspberry Pi single board computers, a gigabit switch and a PC. Raspberry Pi is used to obtain RGB-D frames from each Kinect and send the data to the PC through the LAN using User Datagram Protocol. The PC is used to receive and process the RGB-D frames. Both the depth and color frames were set to a size of 640×480; the values of a single pixel in the depth frame and color frame have a size of 2 bytes and 3 bytes, respectively. Thus, the size of the RGB-D frames from all twelve Kinects is 17.58 million bytes, which can be sent to the PC through the gigabit switch at 7 fps. This frame rate can be increased by decreasing the size of RGB-D frames from Kinect, such as 28 fps when the frame size is set to 320×240.

To evaluate the accuracy of the proposed extrinsic calibration method, we used a motion capture system to obtain the ground truth relative poses between the cameras. The motion capture system requires at least three reflective markers to track the pose of a rigid body, such as the Kinect and the chessboard in our experiments. We attached four reflective markers to both the Kinect and the chessboard (see Figure 5). We placed four markers on the outer corners of the chessboard (see Figure 5b) such that the relative poses between the chessboard and the motion capture system and the Kinect color camera were known. The motion capture system tracked the poses of the markers that were attached to the Kinect to determine the poses of the Kinect color camera. For convenience, we only attached markers to one Kinect and placed this Kinect at twelve different positions in the camera rig to capture RGB-D frames. The accuracy of extrinsic calibration was evaluated using twelve pairs of RGB-D frames from the captured twelve RGB-D frames. The intrinsic parameter of the Kinect color camera is obtained through the Kinect SDK, the depth frame is also aligned with the color image efficiently using an interface provides by the Kinect SDK. The depth maps and color images from Kinects are not undistorted as the distortion is little for a camera with a relatively small FoV of 43${}^{\circ }$. We also compared the pose estimation error with and without undistorting the depth map and color image in Section 4.3.

### 4.2. Evaluation Metrics

Given *M* pairs of 3D point sets $\left\{p_{ij}\right\}$, $\left\{p_{i,j}^{\prime }\right\}$, i=1,…M,j=1,…N, a sequence of relative poses $P_{1},\ldots ,P_{M}\in SE\left(3\right)$ are estimated. Three common metrics are used to evaluate the accuracy of pose estimation through evaluating the alignment error of 3D point clouds and 2D images. The estimated poses are also compared with the ground truth that are obtained through a motion capture system.

#### 4.2.1. Relative 2D Error (R2E)

The relative 2D error is the reprojection error that measures the geometric error corresponding to the image distance. Let *K* denotes the projection matrix, R2E is given by
(6)$$\begin{array}{c} e_{ij}=Kp_{ij}-KP_{i}{p^{\prime }}_{ij},\\ R2E=\frac{1}{MN}\sum_{i=1}^{M}\sum_{j=1}^{N}\sqrt{{\left(e_{ij}.x\right)}^{2}+{\left(e_{ij}.y\right)}^{2}}, \end{array}$$ where $e_{ij}.x$ and $e_{ij}.y$ represent the pixel error of *x*-coordinate and *y*-coordinate respectively.

#### 4.2.2. Relative 3D Error (R3E)

The relative 3D error measures the alignment accuracy of each pairs of 3D point sets after calibration. R3E is calculated as
(7)$$\begin{array}{c} e_{ij}=p_{ij}-P_{i}{p^{\prime }}_{ij},\\ R3E=\frac{1}{MN}\sum_{i=1}^{M}\sum_{j=1}^{N}\sqrt{{\left(e_{ij}.x\right)}^{2}+{\left(e_{ij}.y\right)}^{2}+{\left(e_{ij}.z\right)}^{2}}, \end{array}$$ where $e_{ij}.x$, $e_{ij}.y$ and $e_{ij}.z$ represent the 3D error of *x*-coordinate, *y*-coordinate and *z*-coordinate respectively.

#### 4.2.3. Accumulated 3D Error (A3E)

In a camera system with multiple RGB-D cameras, take the coordinate system of one RGB-D camera as the reference, the pose estimation error will be accumulated while constructing the whole 3D point cloud. For each pair of 3D point sets from camera *i*, i+1, (when *i* equals *M*, i+1 is set to 1), the accumulated 3D error is defined as follows:(8)$$\begin{array}{c} e_{ij}=P_{1}\ldots P_{i}p_{ij}-P_{1}\ldots P_{i+1}{p^{\prime }}_{ij},\\ A3E=\frac{1}{MN}\sum_{i=1}^{M}\sum_{j=1}^{N}\sqrt{{\left(e_{ij}.x\right)}^{2}+{\left(e_{ij}.y\right)}^{2}+{\left(e_{ij}.z\right)}^{2}}. \end{array}$$

### 4.3. Results

While extracting the matched keypoints for each pair of RGB-D frames, we increased the distance threshold of the matched keypoints (denoted by dist_thresh) from two times the minimum distance (min_dist) to the maximum distance (ten times the minimum distance on average), increased the re-projection error threshold to increase the number of matched keypoints to analyze the pose estimation error with respect to the number of keypoints correspondences.

The relative 2D error (R2E) and relative 3D error (R3E) with respect to dist_thresh/min_dist and re-projection error are shown in Table 1 and Table 2 respectively. From Table 1 we can see that, the number of matched keypoints before applying the RANSAC-based filtering method are dramatically increased by increasing dist_thresh/min_dist. However, after applying the RANSAC-based filtering method the number of matched keypoints are only slightly increased. The R2E and R3E are also slightly increased while dist_thresh/min_dist is increased. It can be seen from Table 2 that, increasing the re-projection error threshold will select more matched keypoints, R2E and R3E will also be increased. When dist_thresh/min_dist is set to 2.0, re-projection error threshold is set to 0.1, the optimum value of R2E and R3E are 1.0 pixel and 4.0 mm respectively (see Table 3). When the color image and depth map are undistorted, optimum value of R2E and R3E are 1.4 pixel and 3.8 mm respectively which show slight difference. Because the depth and color camera have a small FoV (43${}^{\circ }$), the distortion of depth map and color image is little. When the SURF detector is used, R2E and R3E are 1.14 pixel, and 4.41 mm respectively.

When the estimated poses are compared with the ground truth obtained through the motion capture system, our proposed calibration method resulted in a rotation error of 0.56 degree and a translation error of 1.80 cm. As to the accumulated 3D error, it is decreased from 100 mm to 33 mm after applying the pose graph optimization. The aligned point clouds before and after pose graph optimization with loop closure can also be observed in Figure 6. It can be seen in the red circle on the bottom of Figure 6a that the point clouds were misaligned; however, these point clouds aligned well in Figure 6b with optimized poses with the loop closure constraint.

The whole calibration process costs average 666 ms to calibrate a pair of RGB-D cameras, most of which was spent on extracting and matching feature points. When the SURF detector is used, it costs 109 ms to calibrate a pair of RGB-D cameras. The method is tested on a computer with a 3.60 GHz CPU.

In comparison, Fernández-Moral et al. proposed to calibrate a similar panoramic RGB-D camera setup with plane patterns, they got a ICP residual (relative 3D error (R3E)) of 2.9 mm [9]. However, the setup was required to be moved around to take about 200 images in order to extract enough planes for all camera pairs to estimated poses, which took too much time. Other approaches that are based on per-camera odometry [15,16] requires to generate useful trajectory for each camera which costs even much more time.

Note that, when the proposed method is applied in the practical usage, the interference between neighboring Kinects that will introduce more noises to the overlapping regions in the depth maps should be avoided. Take the calibration of the setup in Figure 2a as an example, the depth maps and color images from Kinects with sequence numbers 1,3,5,7,9,11 (the twelve Kinects are assigned to sequence numbers from 1 to 12 in order) can be firstly captured and then these Kinects are closed and source images from other Kinects are captured.

The operating range of the depth camera in Kinect v1 is between 0.5 m to 5.0 m [22], a single panoramic RGB-D frame from the panoramic RGB-D camera setup can reconstruct scenes within the circle with a radius of 5 m. Figure 7 presents the reconstruction result of a bedroom (3.3 m × 3 m) and a living room (9 m × 3.5 m) using only one output panoramic frame from the system. The proposed panoramic RGB-D camera setup together with the distributed capturing system provide a 360${}^{\circ }$ FoV to perceive the color and depth information of surrounding things in real time. Compared with typical LIDAR systems, the proposed panoramic RGB-D camera system can provide a much denser point cloud with color information (see Figure 7) which makes it more suitable for some applications in indoor such as 3D reconstruction; mapping; navigation, path planning and avoid obstacles for robots etc.

The panoramic RGB-D camera system provides a 360${}^{\circ }$ FoV, leading to better constraints for localization and can be potentially used to reduce localization and mapping errors.

Our next step is to investigate direct registration methods such as Refs. [23,24], which do not depend on time-consuming keypoint detectors or descriptors for large scale SLAM, using this panoramic 3D vision system.

## 5. Conclusions

In this paper, a new method that relies on well-matched keypoints provided by a feature descriptor-based calibration pattern was proposed to calibrate the extrinsic parameters of the RGB-D cameras in the system. A LAN-based distributed system was developed, which enabled the system to provide panoramic RGB-D frames in real time. The reconstruction of indoor scenes was efficiently and conveniently performed using the panoramic RGB-D 3D vision system. The experiments validated the accuracy and efficiency of the proposed calibration method and the efficiency of the panoramic RGB-D camera setup in 3D reconstruction, and quantitatively demonstrated a higher speed and competitive accuracy compared with existing methods.

## Figures and Tables

**Figure 1 sensors-19-00349-f001:**
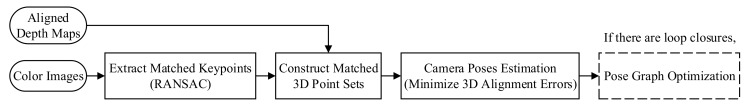
Flow chart of the proposed calibration method.

**Figure 2 sensors-19-00349-f002:**
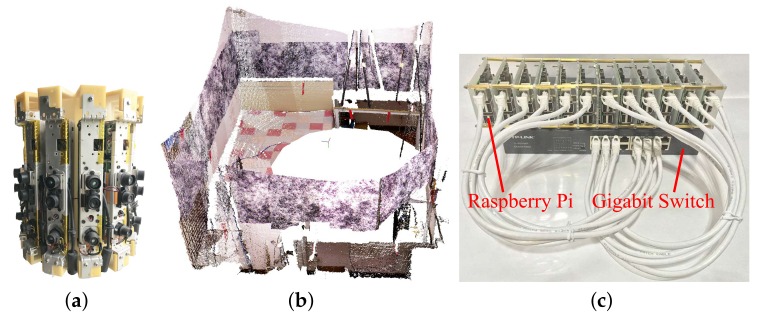
(**a**) The setup used in this study for evaluating the proposed calibration method composed of 12 Kinect v1 RGB-D cameras. (**b**) The panoramic 3D color point cloud obtained by the panoramic 3D vision system. Descriptor-based calibration patterns are pasted on the walls to calibrate multiple RGB-D cameras. (**c**) Distributed capturing system consisting of a gigabit switch and 12 low-cost Raspberry Pi single board computers.

**Figure 3 sensors-19-00349-f003:**
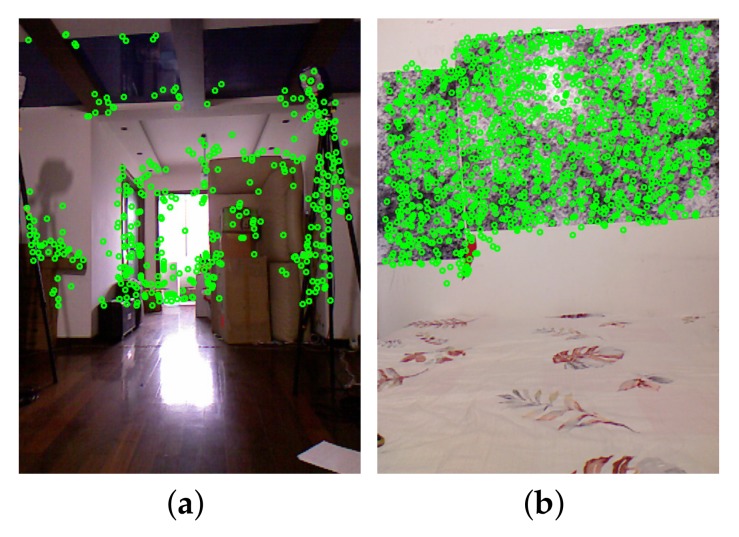
The images depicted in (**a**,**b**) show the detected keypoints of the image captured in a natural scene, and the image captured in a scene with descriptor-based patterns respectively. Keypoints are represented by green dots.

**Figure 4 sensors-19-00349-f004:**
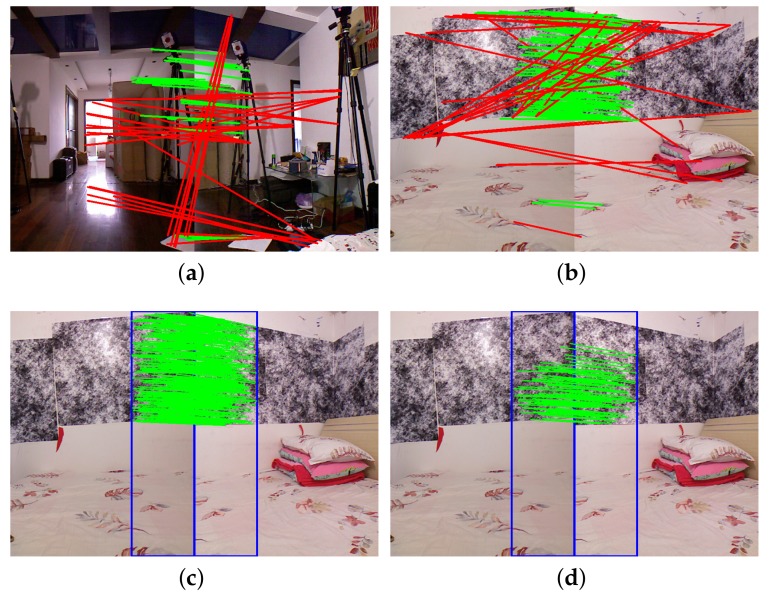
(**a**) Shows the matched keypoints of the image pairs captured in a natural scene. (**b**–**d**) shows the matched keypoints of the image pairs captured in a scene with descriptor-based calibration patterns. The keypoints are detected in the whole image in (**b**), and in the image regions within blue rectangles in (**c**,**d**). The matched keypoints in (**a**–**c**) are accepted if its descriptor distance is below a threshold, (**d**) shows the matched keypoints when the RANSAC-based method is applied to refine the matched keypoints in (**c**). Well-matched keypoints are connected with green lines, poorly matched keypoints are connected with red lines.

**Figure 5 sensors-19-00349-f005:**
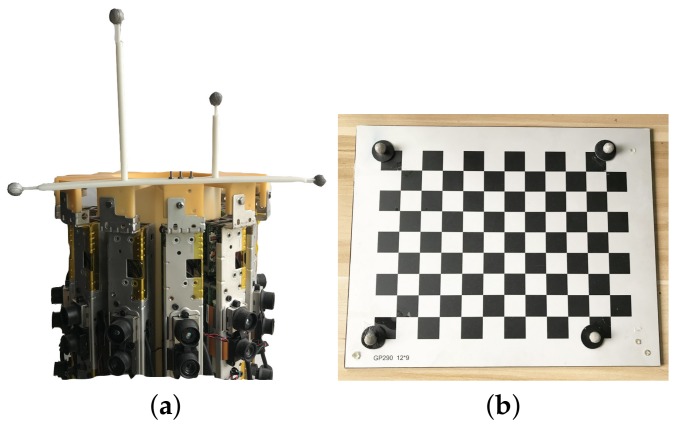
(**a**) Kinect with reflective markers. (**b**) Chessboard with reflective markers.

**Figure 6 sensors-19-00349-f006:**
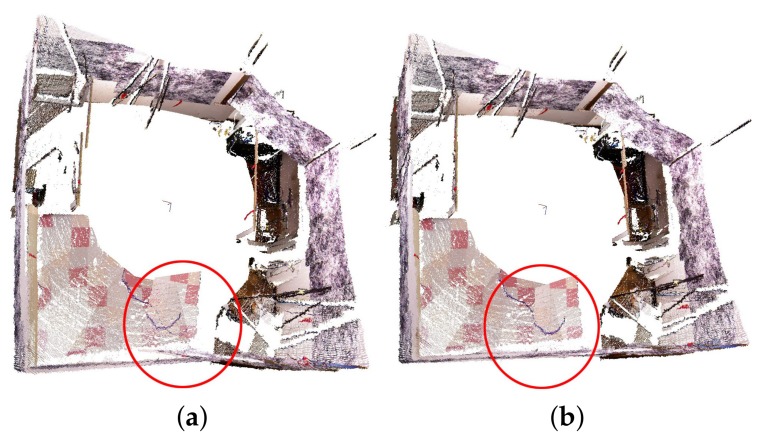
The aligned point clouds before (**a**) and after (**b**) closing the loop. A misalignment can be observed in the red circle on the bottom of (**a**). After pose graph optimization with loop closure, the misalignment is resolved.

**Figure 7 sensors-19-00349-f007:**
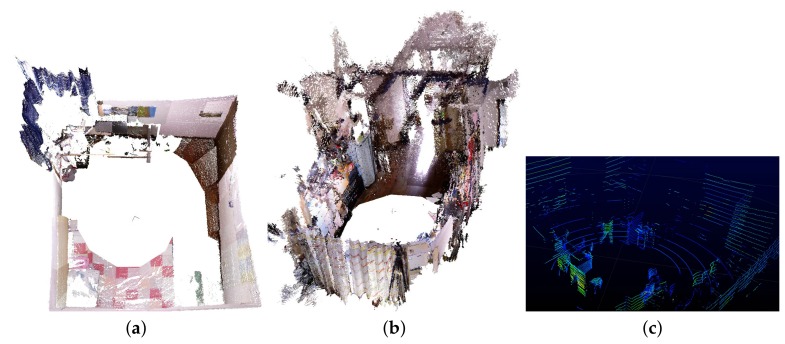
Reconstruction of a bedroom (3.3 m × 3 m) (**a**), and a living room (9 m × 3.5 m) (**b**) using only one output frame from the proposed panoramic 3D vision system. (**c**) Screenshot of the indoor point cloud generated from Velodyne VLP 16 (https://www.youtube.com/watch?v=Se5U2ne5eLk).

**Table 1 sensors-19-00349-t001:** Relative 2D error (R2E) and Relative 3D error (R3E) of estimated poses with respect to the increasing distance threshold of matched keypoints. The re-projection error threshold in RANSAC is set to 0.1 pixel. The minimum errors of R2E and R3E are in bold font.

dist_thresh/min_dist	2	3	4	5	6	7	8	9	10
No. of matched keypoints before RANSAC	62	235	416	652	1064	1316	1379	1464	1535
No. of matched keypoints after RANSAC	11	19	21	22	21	22	21	21	21
R2E (pixel)	1.10	1.08	**1.07**	**1.07**	1.19	1.17	1.17	1.17	1.15
R3E (mm)	**4.01**	5.05	4.67	5.58	4.96	5.11	5.32	5.21	5.26

**Table 2 sensors-19-00349-t002:** Relative 2D error (R2E) and Relative 3D error (R3E) of estimated poses with respect to the increasing re-projection threshold in RANSAC-based keypoints refinement method. dist_thresh/min_dist is set to 2.0. The minimum errors of R2E and R3E are in bold font.

Re-Projection Error Threshold	0.1	0.2	0.5	1.0	1.5	2.0
No. of matched keypoints before RANSAC	62	62	62	62	62	62
No. of matched keypoints after RANSAC	11	19	38	47	52	52
R2E (pixel)	1.10	**0.96**	1.02	1.03	1.06	1.06
R3E (mm)	**4.01**	4.68	5.03	5.26	5.33	5.33

**Table 3 sensors-19-00349-t003:** Relative 2D error (R2E) and Relative 3D error (R3E) of estimated poses of 12 pairs of RGB-D cameras in Figure 2a when dist_thresh/min_dist is set to 2.0, re-projection error is set to 0.1 pixel. “Before” and “After” indicate the number of matched keypoints before and after applying the RANSAC-based filtering method respectively.

Index	Before	After	R2E (pixel)	R3E (mm)
1	83	13	1.23	8.6
2	119	18	0.84	7.7
3	32	7	0.78	2.1
4	44	12	0.94	5.1
5	88	12	0.92	5.0
6	27	9	0.79	2.6
7	65	10	1.05	1.4
8	70	3	2.39	0.8
9	80	12	1.01	3.5
10	57	12	0.95	2.9
11	12	5	1.27	4.1
12	64	16	0.98	3.8
Average	62	11	1.1	3.97

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
