# Peer review of "A Novel Method for Extrinsic Calibration of Multiple RGB-D Cameras Using Descriptor-Based Patterns"

_sensors, 2019, doi:10.3390/s19020349_

Reviewer 1 Report

This paper presents and evaluates a multi-camera simulation of a panoramic RGBD imaging set-up by implementing a method for its automatic calibration using texture added to the scene. The usefulness and novel aspects of the contributed concept are efficiently documented in the text; this basically warrants its publication. In my view, however, certain aspects remain obscure and it is mandatory to clarify and/or discuss them.

1. Besides to a reference (Li et al., 2014), the authors should explain in more detail exactly what is a "descriptor-based pattern" and, especially, how one selects a calibration pattern, i.e. whether it has to satisfy some particular specifications or it is a pattern of random texture, whether it necessarily has to cover the whole 360-degree space to appear on all images of the imaging system, etc. Such qualifications are needed for the reader to grasp the requirements and limitations of the proposed approach, its functionality and ease of use (see also comment no. 7 below).

2. The authors state that the matching results are produced using a descriptor matcher, implemented in OpenCV, where a matched keypoint is accepted if its descriptor distance is below a threshold (72); they claim that all poorly matched keypoints are thus filtered out (78). But it is well-known that simply matching descriptor vectors is generally not enough for avoiding false matches; this is also observed in the results where increase of the distance threshold of matches increases the errors (140). As a rule, therefore, outliers are removed using appropriate rigorous geometric constraints (e.g. the epipolar constraint) and RANSAC-type robust estimators. Apparently, no such constraint has been used here. The cameras of the panoramic system are of course close to each other; hence the fundamental matrices are expected to be rather weakly estimated - still their use will probably filter out more faulse matches. The authors should consider and discuss this rather obvious alternative instead of relying on mere descriptor matching.

3. In this work the keypoint detector is only applied to the overlapping regions of neighboring images (76). How are these regions identified, are they given manually or automatically selected using some rough knowledge of the imaging geometry?

4.  The authors say that due to the different spatial positions and intrinsic parameters of depth and color cameras in an RGB-D camera, the depth map is not aligned with the color image. The extrinsic parameters between the depth and color camera are used to align the depth map with the color image (82). This means, as I understand it, that the intrinsic camera parameters of both cameras are assumed to be known. How are they known? And: do the authors assume that all individual depth cameras (and, respectively, all color cameras) of their panoramic system have the same intrinsic parameters? In their evaluation a single RGBD camera has been used, yet the question obviously matters in real practical cases. And, finally, what about radial lens distortion? Such issues must be addressed and made clear to the reader.

5. In section 3.2 it is not clear to the reader which are the two 3D point sets (p, p') whose differences are minimized. The authors start with (u,v,Z) of an extracted point on the depth camera, calculate its X,Y coordinates (Eq. 1), transform it through T to (X',Y',Z') and back-project it onto the color camera to obtain (u',v',Z'). Rather than minimizing the back-projection image error, they say that they minimize the differences of two 3D point pairs p, p' (Eq. 4). Which are these? It should be clearly stated exactly which are the elements of the two vectors p, p' and what they represent.

6. Next, the authors proceed to solve for the relative poses of adjacent RGBD cameras. Is it assumed that the above-mentioned poses between depth and color camera of individual RGBD cameras have been determined beforehand, or are they part of a unified adjustment as unknowns? And do the authors take for granted that all RGBD cameras share an identical pose between their depth and color camera? Those question should also be clarified.

7. Overall, the proposed approach uses a single acquisition by the RGBD system at the price of adding ubiquitous descriptor-base patterns, which is mainly attributed to narrow FoV and hence limited overlap. Do the authors think that they might have avoided altogether the tedious addition of extra texture to the scene by simply using more imaging positions and performing a global bundle adjustment by exploiting the multiple overlaps (with each additional imaging location introducing just 6 new unknowns)?

Author Response

Dear Editor and Reviewers,

We thank you for reviewing our manuscript. We value your insightful comments and have revised the manuscript accordingly. Every comment has been carefully addressed, as the attached PDF file.

Reviewer 2 Report

The problem of Extrinsic Calibration of Multiple RGB-D Cameras is very important to computer vision applications, chiefly robotics and augmented reality for example.

The authors placed correctly the proposed method but failed when comparing the presented research to the state-of-the-art. Does the proposed research out-performs  the existing ones in therms of the error obtained? computational simplicity? application in real-time? other theoretical novelty? The authors must clear all the previous itens in the revised version.

Section 2, should be more  clear in the contribution of the authors. In which block is the novelty? and why? In the text is not clear if the authors proposed any kind on new contribution.

Again in section 3.1 , 3.2 and 3.3 are presented well known methods from the literature.

In section 4, experimental results are presented which show good results, but a mandatory comparison to state of the art methods is needed, specially the ones identified by the authors in section 1.

The authors should discuss the application of the proposed system in real world applications, and why should be used such a system related to the nowadays cheaper velodyne type LIDAR systems. what is the sample time of the overall system, i.e., how much time is the 360 field of view image is updated to be used by a robot, for example.

Author Response

(The authors gave the same response as above.)
